# The Role of Neutrophils and Neutrophil Extracellular Traps in Vascular Damage in Systemic Lupus Erythematosus

**DOI:** 10.3390/jcm8091325

**Published:** 2019-08-28

**Authors:** Liam J. O’Neil, Mariana J. Kaplan, Carmelo Carmona-Rivera

**Affiliations:** Systemic Autoimmunity Branch, National Institute of Arthritis and Musculoskeletal and Skin Diseases, National Institutes of Health, Bethesda, MD 20892, USA

**Keywords:** neutrophils, atherosclerosis, interferon, vascular damage

## Abstract

Systemic lupus erythematosus (SLE) is an autoimmune syndrome of unknown etiology, characterized by multi-organ inflammation and clinical heterogeneity. SLE affects mostly women and is associated with a high risk of cardiovascular disease. As the therapeutic management of SLE improved, a pattern of early atherosclerotic disease became one of the hallmarks of late disease morbidity and mortality. Neutrophils emerged as important players in SLE pathogenesis and they are associated with increased risk of developing atherosclerotic disease and vascular damage. Enhanced neutrophil extracellular trap (NET) formation was linked to vasculopathy in both SLE and non-SLE subjects and may promote enhanced coronary plaque formation and lipoprotein dysregulation. Foundational work provided insight into the complex relationship between NETs and immune and tissue resident cells within the diseased artery. In this review, we highlight the mechanistic link between neutrophils, NETs, and atherosclerosis within the context of both SLE and non-SLE subjects. We aim to identify actionable pathways that will drive future research toward translational therapeutics, with the ultimate goal of preventing early morbidity and mortality in SLE.

## 1. Introduction

Atherosclerosis, a common cause of morbidity and mortality in the general population [1], continues to be a hotbed of translational and clinical research. Once thought to be simply the result of vascular accumulation of lipids, the understanding of atherosclerosis greatly evolved over time. Inflammation plays a fundamental role in the evolution of this disease process [2]. Consistent with this, patients with various autoimmune diseases have a clear predisposition toward the development of atherosclerosis [3]. This association is particularly striking in systemic lupus erythematosus (SLE), where observational and epidemiologic studies described up to 50-fold increased risk of developing cardiovascular disease compared with non-SLE patients [4,5]. In recent years, neutrophils were implicated in the inflammation observed in the pathogenesis of both athero-embolic disease and SLE pathophysiology [6]. Understanding how neutrophils mediate atherosclerosis may provide an understanding of actionable pathways in both SLE and non-SLE subjects and, ultimately, could contribute to improving patient management and clinical outcomes. In this review, we provide an overview of SLE and atherosclerosis, discuss the role of neutrophils and neutrophil extracellular traps (NETs) in atherosclerosis, and provide SLE-specific evidence of NETs mediating atherosclerotic disease.

## 2. SLE and Atherosclerosis: Epidemiology

SLE is a systemic autoimmune syndrome with profound disease heterogeneity, characterized by dysregulated innate and adaptive immune responses and the presence of a type 1 interferon signature in a significant number of patients. Individuals affected by SLE also have enhanced cardiovascular risk due to the development of premature atherosclerosis, although the magnitude of the risk reported in the literature varies widely [7]. Indeed, the elevated risk of atherosclerosis in SLE ranges from 10–50-fold, while the risk of myocardial infarction is 5–8-fold. Women with SLE aged 45 to 54 appear to have the greatest fold difference from their healthy counterparts after controlling for traditional risk factors [8,9]. SLE has a bimodal pattern of mortality, and atherosclerosis is likely the most important cause of late death in these patients [10]. Similar to most associative findings in SLE, disease heterogeneity clouds definitive conclusions from well-designed epidemiological studies, and atherosclerosis is no exception. Disease activity [11], disease duration [12], clinical damage scores [13], and renal disease [14] are all linked to the development of atherosclerosis. Traditional Framingham cardiac risk factors (hypertension, dyslipidemia, smoking, etc.) are all associated with atherosclerosis in SLE, but cannot fully explain the increased risk. Indeed, immune dysregulation characteristic of SLE appears to play a key role in driving premature atherogenesis. What remains evident is that atherosclerosis represents an important factor for SLE morbidity and mortality, and understanding the inter-relations between atherosclerosis, autoimmunity, and systemic inflammation may uncover key actionable pathways for improvement in patient outcomes.

### Lupus-Specific Mechanisms of Enhanced Atherosclerosis

Standardized prediction models underestimate the incidence of cardiovascular events in SLE patients, suggesting that there are unique pathways that contribute to the development of early atherosclerosis. Dysfunctional high-density lipoprotein (HDL), a proinflammatory biomarker for atherosclerosis, is elevated in SLE patients [15]. SLE patients also have antibodies against apolipoprotein A1 (ApoA1) and HDL, which may neutralize their atheroprotective effects [16]. Invariant natural killer T cells (iNKTs) respond to lipid antigens and modulate the inflammatory response observed in autoimmune atherosclerosis [17]. SLE patients with atherosclerosis have increased numbers of iNKT cells which likely have a protective role. However, iNKT functionality is altered in patients who transition from subclinical to overt cardiovascular disease—suggesting their role in the establishment of clinical disease [18]. Elevated levels of homocysteine may also play an important role in SLE-related premature atherosclerosis. Homocysteine levels are increased in lupus [19], which has adverse effects on smooth muscle cells and platelet activation [20]. Lupus is strongly associated with the expression of type 1 interferon-regulated genes [21], and this pathway likely also contributes to atherosclerotic risk [22]. Type 1 interferon modulates macrophages and cytotoxic T cells to promote atherosclerotic plaque formation. Finally, a clear reduction in atheroprotective factors in SLE, such as endothelial progenitor cells [23] and immunoglobulin M (IgM) [24], also likely plays an important role in the development of disease.

## 3. Inflammation, Neutrophil Extracellular Traps (NETs), and Atherosclerosis

The landmark study associating systemic elevation of C-reactive protein (CRP) and the development of cardiovascular disease was the first major insight into the importance of inflammation in atherosclerosis [25]. Replication of these findings [26], and the clinical efficacy of canakinumab, a monoclonal antibody therapy that neutralizes the pro-inflammatory cytokine interleukin (IL)-1β [27], provided clarity on the important role that inflammation plays in the development of atherosclerosis. However, the complex pathophysiology of atherosclerosis and the heterogeneity of patient presentation made identifying other inflammatory markers a more difficult task. It appears that both humoral and cellular immunity contribute in meaningful ways to atherogenesis at different stages of this process [28]. Given the abundance of neutrophils and their armamentarium of cytotoxic mediators, it is clear that they have the capacity to substantially harm tissue and promote inflammatory responses in a diseased artery [29]. Indeed, research over the last decade implicated neutrophils in the development of atherosclerosis [30].

In humans, neutrophils are the most abundant leukocyte in circulation. These sentinel cells are short-lived (although they may live longer in inflammatory environments [31]), and they are well equipped to kill infiltrating pathogens [32]. They possess a broad spectrum of cytotoxic mediators and respond to invading microbes using phagocytosis, generation of reactive oxygen species, and degranulation [33]. A unique feature of neutrophils is that they synthesize their granules prior to their release from the bone marrow into circulation [34]. This allows a rapid and robust response to pathogens without the need for transcriptional machinery. Perhaps the most distinctive cytotoxic function of neutrophils is their ability to undergo a distinct form of cell death that leads to the release of neutrophil extracellular traps (NETs). Vital NETosis is an alternate form of DNA release that does not result in cell death, whereby a neutrophil maintains its phagocytic function despite compromise of its cellular membrane [35]. During most, but not all forms of NET formation, peptidylarginine deiminase-4 (PAD4) enzymatically converts arginine into citrulline on histone tails, promoting chromatin decondensation in the nucleus [36]. Neutrophil elastase (NE) and myeloperoxidase (MPO) are translocated into the nucleus and further promote chromatin release and nuclear membrane dissolution. In later stages, the plasma membrane is disrupted, and chromatin decorated with granule proteins is released in web-like structures to the extracellular space. It is thought that chromatin is used by the host immune system to entrap microbes and expose them to cytotoxic compounds such as proteases [37]. Importantly, during some forms of NET formation, mitochondrial DNA may also be incorporated into these structures, which adds to their pathogen-fighting abilities [38].

Dysregulated NET formation was recently implicated in a variety of inflammatory conditions including autoimmunity, autoinflammation, thrombosis, malignancy, and sepsis [39,40]. Intriguingly, NETs are proposed to play a fundamental role in SLE pathophysiology and to be a primary driver of increased vasculopathy in these patients. Enhanced NET formation and impaired NET clearance are described in peripheral blood and various tissues as hallmarks of SLE [41]. Reduced NET clearance is partially explained by impaired DNAse-1 activity [41], the key enzyme for NET degradation. Neutrophils respond to, and produce [42] type-1 interferon, a key cytokine in SLE pathogenesis. Furthermore, subsets of lupus neutrophils readily undergo enhanced spontaneous NET formation [43]. NETs may also have immunoregulatory properties; for example, NETs may reduce the release of proinflammatory cytokines in macrophages [44]. Unraveling the connection between SLE, NET formation, and atherosclerosis may allow for the discovery of new, actionable pathways that are relevant to not only SLE, but also to the general population.

## 4. Neutrophils Facilitate Atherosclerosis Development and Progression

The process of atherosclerosis generally evolves over many years, and its initiation is a quiescent, subclinical event. Selectins are proteins which allow circulating neutrophils to slow, roll, and tether to the endothelium [45]. These cell adhesion molecules are expressed by endothelial cells as the initiating event in atherosclerosis, particularly in an environment induced by dyslipidemia. Indeed, in mice that are fed high-fat diets, neutrophil accumulation within the arteries [46] is a very early event. C–C motif chemokine ligand 5 (CCL5) allows neutrophils to firmly attach to the endothelium, where it begins to promote pathogenic changes to initiate atherogenesis. Granule proteins and radicals promote endothelial dysfunction [47] and monocyte recruitment [48], as well as increase vascular permeability [49]. The recruitment of monocytes is of particular importance, as these cells provide the precursors to endothelial macrophages, which leads to the development of proinflammatory M1 cells and foam cells [50,51,52], the latter of which are crucial for the development of plaque. Finally, plaques that form over several years to decades have the propensity to rupture or destabilize, resulting in thrombus formation. Neutrophils express a variety of proteases such as matrix metalloproteinases or serine proteases. These enzymes appear to be important in the development of intraplaque hemorrhage [53] and degradation of the fibrous cap [54].

Activated neutrophils are the predominant cell type found within the thrombus of patients having a coronary event [55]. Furthermore, NETs within the thrombus significantly correlate with cardiac infarct size and, inversely, with electrocardiogram (EKG) ST-segment resolution. These findings implicate NETs in the clinical severity of a coronary lesion. Thrombus specimens following percutaneous coronary intervention were found to be infiltrated with neutrophils, with a subset of lesions containing granule–DNA complexes, suggestive of NETs [56]. Moreover, granule–DNA complexes are also found in coronary plaque erosions and intra-plaque hemorrhages, and they are not only detectable in the interstitial adventitia but also in the lumen of microvessels [57]. NETs were also found shown in monocyte-depleted Lysm^egfp/egfp^ atherosclerosis-prone apolipoprotein-E^−/−^ mice within the carotid bifurcation using intravital microscopy [58]. High-mobility group box 1 (HMGB1) synthesized by activated platelets was described as an inducer of NET formation by coronary artery neutrophils (Figure 1) [59]. Cholesterol crystals, known to play an important role in plaque formation, also induce NET formation. The interplay between NETs, cholesterol crystals, and macrophages can promote pro-inflammatory cytokine release such as IL-1β [60]. NETs isolated from coronary thrombectomy specimens also appear to be coated with IL-17A and IL-17F, which may contribute to inflammation, platelet aggregation, and thrombus expansion [61].

### 4.1. The Role of Serine Proteases

Apolipoprotein-E^−/−^ and the low-density lipoprotein (LDL)^−/−^ mice are important models for studying the pathogenesis of atherosclerosis. Deletion of neutrophil elastase (NE) and proteinase-3 (PR3) in the apolipoprotein-E^−/−^ mouse leads to reduced NET formation, along with a reduction in atherosclerotic lesions and systemic inflammation [60]. In the same mouse model, NETs within atherosclerotic plaques can stimulate plasmacytoid dendritic cells (pDCs) to produce type 1 interferons and contribute to the development of plaque [65]. Indeed, depletion of pDCs leads to a substantial reduction in plaque formation in this model, highlighting an important role for type 1 interferons in plaque formation. A triple knockout mouse model (ApoE/PR3/NE KO) displays substantially reduced atherosclerotic plaque formation as well, a finding that appears to be at least in part mediated by the intact DNA within NETs. In a separate study, loss of NE alone did not affect lesional formation without the concurrent loss of PR3 [67]. Neutrophil serine proteases (NE and cathepsin G), found in high quantity in NETs [68], were described as important mediators of coagulation. In a coagulation murine model, elastase (ELANE)^−/−^/Cathepsin G (CTSG)^−/−^ mice have impaired coagulation when compared to wild-type mice, with more fragile and smaller thrombi, along with heightened bleeding times [66]. NE was described as a predominant protease in clotting mechanisms, with near complete recovery of normal phenotype in this mouse model after injection of recombinant NE. NE cleaves tissue factor pathway inhibitor (TFPI), a key anticoagulant found in platelets. NETs are required for the co-localization of TFPI and NE, as this effect is ameliorated by treatment with DNase-1, a nuclease that disrupts NET architecture, both in vitro and in vivo.

### 4.2. Cellular Toxicity by Histone H4 and Matrix Metalloproteinases

NETs may promote arterial dysfunction by directly interacting with endothelial cells. In ApoE^−/−^ mice, neutrophil depletion resulted in increased smooth muscle cells (SMC), while neutrophilic mice generated by C–X–C motif chemokine receptor 4 (CXCR4) deletion had heightened SMC death [62], suggesting that neutrophils regulate SMC survival. In this study, authors also reported that C–C motif chemokine ligand 7 (CCL7) synthesized by SMC enhances NET formation that then triggers SMC cell death. Histone H4 contained in NETs is a key mediator in the induction of SMC cell death through the formation of cell membrane pores and membrane destabilization. The use of histone inhibitory protein (HIP) in mice with pre-existing atherosclerotic lesions increased the stability of these lesions without altering neutrophil recruitment. In conjunction with this mechanism, matrix metalloproteinases (MMPs, specifically MMP9) within NETs induce endothelial cell cytotoxicity and promote dysfunctional vascular relaxation [63]. Importantly, this is enhanced when assessing NETs derived from lupus neutrophils. Histones can also induce endothelial dysfunction [69]; however, whether histone modification alters the capacity to damage the endothelium remains to be systematically examined. Whether or not MMPs can promote SMC death remains to be determined, but there may be synergism between NET molecules that drives artery-specific cellular toxicity. 

### 4.3. PAD-4, a Potential Target for Therapy

PAD4, one of the enzymes that mediates the citrullination of arginine, appears to play an important role in NET formation following certain types of stimulation [70]. Although the role of PAD4 in the clinical and immunological manifestations of murine SLE is debated [71,72], studies investigating the role of this molecule in NET formation and atherosclerosis are much clearer. Inhibition of PAD4 by the pan-PAD inhibitor Cl-amidine in ApoE^−/−^ mice significantly reduced atherosclerotic lesion size and prolonged time to thrombosis in a photochemical injury model [73]. This model confirmed enhanced systemic NET formation in murine atherosclerosis and a type 1 interferon response within the aortic tissue, all of which were mitigated by Cl-amidine therapy. These findings were recapitulated in a PAD4^−/−^ myocardial infarction mouse model, which displayed improved ejection fraction percentage and reduced infarct size compared to wild-type animals, with no additional benefit of DNase I treatment. Furthermore, obesity-induced endothelial dysfunction is also ameliorated by Cl-amidine [74]. PAD4 is also responsible for promoting von Willebrand factor–platelet string formation, which accelerates thrombosis upon endothelial injury in mice [75]. In human carotid specimens, NETs were shown to be more closely associated with erosion-prone rather than rupture-prone pathology. Indeed, there is clear evidence of neutrophil infiltration and NET formation in mice exposed to flow perturbation at sites of intimal expansion [76]. Deficiency of PAD4 in a murine model of endothelial injury revealed reduced NET formation, preservation of endothelial continuity, and smaller intraluminal thrombus. Indeed, protection through a myeloid-specific deletion of PAD4 confirms the importance of NET formation in murine atherosclerosis [77]. Although some debate exists over the role of NET formation in modulating SLE murine manifestations [71,72], PAD4 inhibition in humans could have a variety of consequential effects. Nonetheless, PAD4 may represent an important drug target via which NET-induced atherogenesis could be significantly altered. 

### 4.4. Mitochondrial DNA and Reactive Oxygen Species

Mitochondrial DNA and oxidative stress may play an important role in the pathogenic effect of NETs in atherosclerotic disease [78]. There is correlation between atherosclerosis, aging, and mitochondrial ROS generation [79]. Bone marrow transplantation from mitochondrial catalase transgenic mice into *low density lipoprotein receptor* (Ldlr)^−/−^ mice resulted in reduced lesional mitochondrial oxidative stress, decreased NET formation, and protection from plaque development [80]. This observation was not apparent in younger mice, but rather in an aging mouse model of atherosclerosis. Whether these differential findings in varying age groups occur in humans remains to be determined. 

High-density lipoprotein (HDL)-mediated cholesterol efflux is an important factor that protects from the generation of foam cells and atherosclerosis in the general population and in autoimmunity [81]. Nicotinamide adenine dinucleotide phosphate hydrogen (NADPH) oxidase (NOX) and nitric oxide synthase activity within NETs can induce nitrogen and reactive oxygen species in the extracellular space [82]. NETs can mediate HDL oxidation, which drives mishandling of lipoprotein homeostasis, transforming HDL into a proatherogenic, proinflammatory form that can promote proinflammatory responses in macrophages [64]. Low-density granulocytes (LDGs), a subset of neutrophils found in high abundance in SLE patients [6], may also play an important role in atherogenesis. LDGs have a propensity to form NETs spontaneously in vitro (Figure 2). These NETs also have a higher degree of mitochondrial ROS and oxidized mitochondrial DNA (mtDNA) compared to their normal dense counterparts [38]. Oxidized mitochondrial DNA released from neutrophils can drive type 1 interferon in lupus [38,83]. Furthermore, LDG gene signatures correlate with non-calcified plaque burden and vascular inflammation in SLE patients [84]. Similar findings were also shown in psoriasis, another systemic autoimmune disease [85]. Thus, there may be a link between LDG NETs, mitochondrial ROS, and lipoprotein dysfunction that drives autoimmune-specific heightened atherosclerosis. 

### 4.5. NETs as a Biomarker for Subclinical Atherosclerosis

Despite a clear role for NETs in the pathogenesis of atherosclerosis, the utility of cell-free DNA and related biomarkers still remains relatively understudied. In a cohort of patients with coronary artery disease, extracellular double-stranded DNA (dsDNA), nucleosomes, and MPO–DNA complexes are higher in patients with severe coronary artery disease (CAD) compared to those without CAD. These parameters also correlated with the number of calcified plaques, in vitro thrombin generation, and future major cardiac events [86]. Despite NET complexes being a putative relevant biomarker in SLE [63], it is unclear if these correlate with measures of subclinical atherosclerosis or patient-related outcomes. Furthermore, there remain methodological inconsistencies by which NETs are measured from human specimens. NET proteins bound to DNA (citrullinated histones, MPO, and NE most commonly) are used as a surrogate for measuring NET formation. More broadly, circulating nucleosomes and cell-free DNA are much less specific, as these remnants could be derived from a variety of sources.

## 5. Concluding Remarks

Recent studies emphasized the role of a pathogenic interplay between neutrophils and the type 1 IFN response as important drivers of atherosclerosis and vasculopathy in SLE. While the potential for intervening in these pathways remains an intriguing concept, much work is needed to translate these findings into clinical practice. The role of PAD inhibition in atherosclerosis is an attractive approach but needs to be better investigated in pre-clinical models before being investigated in human trials. Canakinumab therapy reduces athero-embolic disease in non-SLE patients; however, blocking this cytokine is not routine therapy in SLE and may have deleterious consequences given that vasculogenesis in SLE is impaired [87]. Furthermore, CRP, an important inclusion factor for patients in the representative clinical trial, is not a known biomarker of SLE inflammation, and CRP responses may be decreased in this disease due to type 1 IFN effects [88]. Inhibiting the toxic and potentially synergistic effects of histones and MMPs on structural arterial cells may also be an important pathway to target. Future studies will hopefully shed light on this important question in the context of both SLE and non-SLE atherosclerosis. Ultimately, research in SLE atherosclerosis remains in its infancy. There appears to be reason for optimism, as research highlighted in this review suggests there was important progress in the understanding of the pathogenesis of lupus vasculopathy. Carrying this momentum toward translatable clinical solutions remains a lofty, but attainable goal.

## Figures and Tables

**Figure 1 jcm-08-01325-f001:**
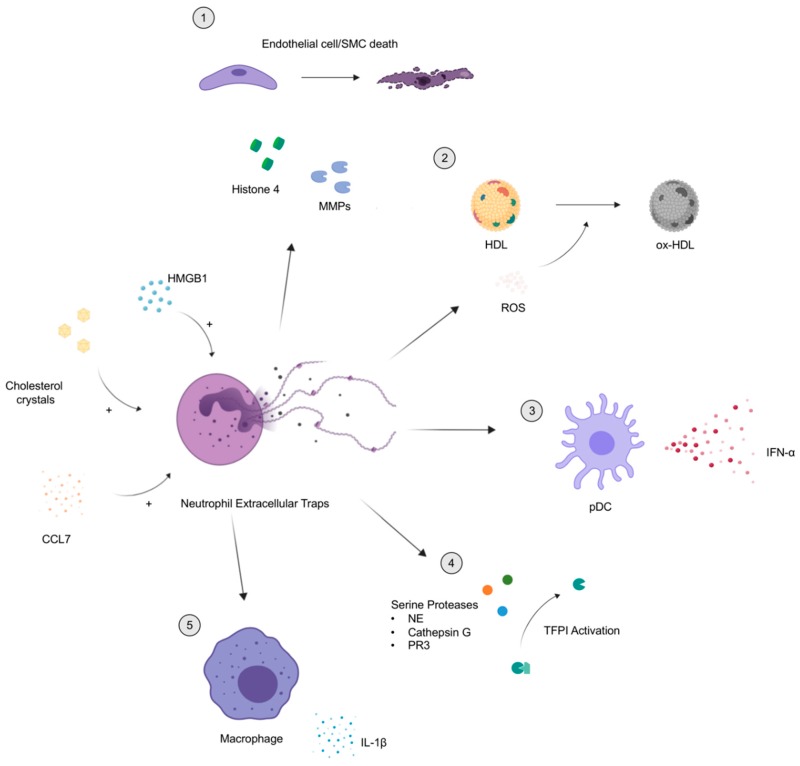
Mechanistic overview of the atherosclerosis pathways mediated by neutrophil extracellular traps (NETs). Molecules such as high-mobility group box 1 (HMGB1) [59], cholesterol crystals [60] and C–C chemokine ligand 7 (CCL7) [62] can exacerbate NET formation. (1) During this event, Histone H4 and matrix metalloproteinases (MMPs) [63] are externalized and induce cell death in smooth muscle cells and endothelial cells (key structural cells in the artery), respectively. (2) Also, reactive oxygen species (ROS) are released during NET formation, which triggers oxidation of high-density lipoprotein (HDL), impaired cholesterol efflux capacity [64], and formation of foam cells. (3) NETs also induce synthesis of interferon alpha (IFN-α) by plasmacytoid dendritic cells [65], a mechanism linked to both systemic lupus erythematosus (SLE) and atherosclerosis. (4) NETs are decorated by serine proteases that were shown to deactivate tissue factor pathway inhibitor (TFPI), an inhibitor of thrombosis [66]. (5) NETs lead to activation of macrophages and, ultimately, the release of interleukin 1 beta (IL-1β) [60], a key inflammatory factor in atherosclerosis.

**Figure 2 jcm-08-01325-f002:**
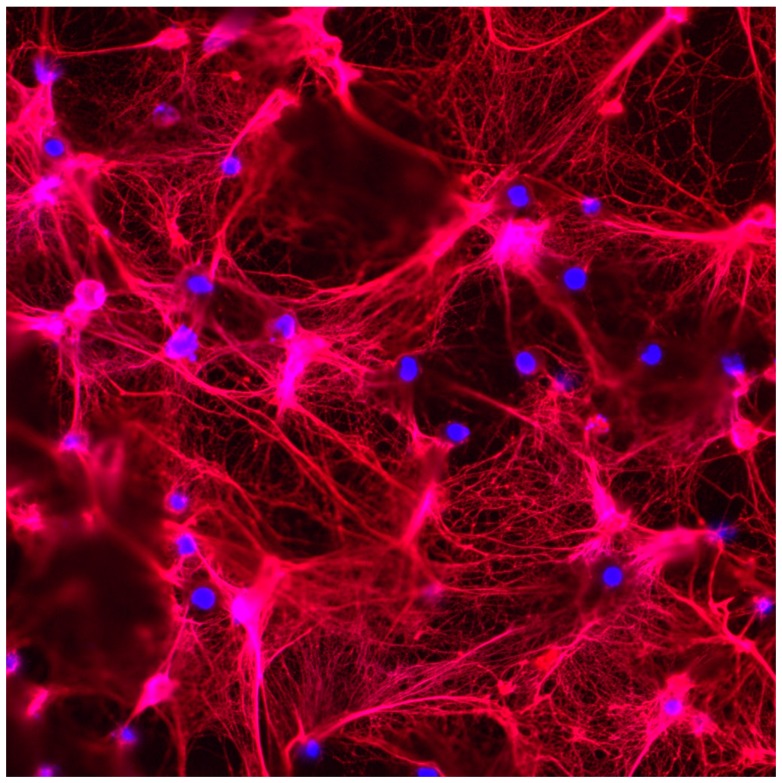
Low-density granulocytes (LDGs) from SLE patients display enhanced NET formation. Immunofluorescence analysis shows that LDGs isolated from a lupus patient have the propensity to form exuberant NETs in vitro. Red is myeloperoxidase (MPO); blue is DNA. Original magnification 400×.

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
