# Peer review of "The Role of Neutrophils and Neutrophil Extracellular Traps in Vascular Damage in Systemic Lupus Erythematosus"

_jcm, 2019, doi:10.3390/jcm8091325_

Round 1
Reviewer 1 Report
This is an interesting, elegant and very well written review analyzing the mechanistic link between neutrophils, NETs and atherosclerosis within the context of both SLE and non-SLE subjects.
My main concerns about its organization and content are:
While there is an extensive analysis of studies evaluating the role of neutrophils and NETs in the development of atherosclerosis in human cardiovascular diseases and animal models, I miss a section dedicated to those studies in the context of SLE (there are included only some referential sentences in different sections). This section should be created, and specific studies performed in the context of SLE with increased CV risk should be reviewed. It should be also illustrative to add a scheme or table including the most relevant findings in this area.
The section ‘PAD-4, a potential target for therapy’should appear as a separate section, not included as a subheading of section 5. Other potential therapeutic targets??? What about the influence of standard therapy on those mechanisms?
Section 4 does not appear in the manuscript. Is it missing or is just a typographical error?
Author Response
This is an interesting, elegant and very well written review analyzing the mechanistic link between neutrophils, NETs and atherosclerosis within the context of both SLE and non-SLE subjects.
My main concerns about its organization and content are:
While there is an extensive analysis of studies evaluating the role of neutrophils and NETs in the development of atherosclerosis in human cardiovascular diseases and animal models, I miss a section dedicated to those studies in the context of SLE (there are included only some referential sentences in different sections). This section should be created, and specific studies performed in the context of SLE with increased CV risk should be reviewed. It should be also illustrative to add a scheme or table including the most relevant findings in this area.
We thank you for the suggestion. We added a section dedicated to lupus-specific mechanism of enhanced atherosclerosis. Please refer to section 2.1 line 60
The section ‘PAD-4, a potential target for therapy’should appear as a separate section, not included as a subheading of section 5. Other potential therapeutic targets??? What about the influence of standard therapy on those mechanisms?
We thank you for the suggestion.
Section 4 does not appear in the manuscript. Is it missing or is just a typographical error?
Indeed, there was a typo, we updated it accordingly. Please refer to line 123
Reviewer 2 Report
This is a well written concise report on the impact of neutrophils in vascular damage of SLE. Relevant aspects are covered. I do not have further comments.
Author Response
This is a well written concise report on the impact of neutrophils in vascular damage of SLE. Relevant aspects are covered. I do not have further comments.
Thank you for your comment!
Reviewer 3 Report
The manuscript is overall well presented, and describes clearly the role of neutrophils and NET in atherosclerosis, however the SLE-specific mechanisms are not detailed (except at the very end of section 5.4).
Specific comments:
Line 71: a reference describing the presence of PMN or NET at sites of inflammation in diseased artery would be helpful.
Line 72: references are needed.
Line 74 : PMN are usually described as short-lived cells. However, recent data show they survive longer than believed. Activated PMN show a longer survival and PMN life span in vivo reaches 5 days in mice (Pillay et al., Blood, 2010, 116:625).
Line 79: different forms of NET formation have been described. Some do not lead to PMN death. Therefore NET formation is not always associated to cell death which is then named NETosis.
Line 80: Likewise, some reports suggest that some forms of NET formation do not require PAD4-mediated citrullination.
Line 85: not only "double-stranded DNA" is decorated and released, as histones are also present in NET, therefore “chromatin” should be mentioned.
Line 86: Likewise “chromatin” and not “DNA strands” entrap microbes.
Line 88: it should be indicated that mitochondrial DNA may only be present in a particular mode of NET formation.
Line 94: reference 27 does not demonstrate enhanced NET formation and impaired clearance in SLE. The reference should be changed or the sentence removed.
Line 95: it should be mentioned that the role of classical NET in SLE has been challenged (Gordon et al., JCI Insight, 2017, 2(10):e92926).
Paragraphs 1 and 3 of section 3 could be more detailed.
In paragraph 3 of section 3: I would mention that neutrophils produce IFN-alpha, a key cytokine in SLE (Lindau et al., 2014, Ann Rheum Dis, 73(12):2199); that there is an impaired clearance of NET in SLE (Hakkim et al., PNAS, 2010, 107(21):9813), probably as a result of low DNase 1 activity, which is supported by data in DNase1-KO mice developing a lupus-like disease (Napirei et al., Nature genetics, 2000, 25:177 and recently confirmed by Kenny et al., Eur J Immunol, 2019, 49:590); and that, however, NET may also have immuno-regulatory properties (Ribon et al., 2019, J Autoimmun, 98:122; Kienhöfer et al., JCI Insight, 2017, 2(10):e92920).
There is no section 4.
Line 104: CCL5 is C-C motif chemokine ligand 5 (“ligand” is missing).
Line 112: remove “are”
Line 115: thrombus NET formation should be explained.
Lines 119 and 121: reference 40 only shows co-staining of DNA and neutrophil elastase (NE), while there is no DNA staining shown in reference 41. Therefore I would be cautious in concluding the structures are NET, especially classical NET. It can be referred to DNA-NE or MPO-citrullinated H3 complexes.
Line 154: the meaning of the sentence is not clear according to results described above.
Line 167: the sentence is confusing, I would remove “lesional”.
Section 5.3: I would modulate conclusions according to the challenged role of PAD4 in lupus (Gordon et al., JCI Insight, 2017, 2(10):e92926) and the potential anti-inflammatory role of NET in lupus (Kienhöfer et al., JCI Insight, 2017, 2(10):e92920).
Section 5.3: it is not explained whether all these PAD4-mediated effects are related to NET formation.
Line 198: the sentence is not clear.
Section 5.4: mitochondrial DNA involvement is mentioned but not detailed.
Section 5.5: It should be explained that when studies aim at analyzing circulating NET, measuring circulating DNA or circulating nucleosomes is not enough to prove these components are NET-derived. This is often confusing in the literature and leads to over-interpretation.
Author Response
The manuscript is overall well presented, and describes clearly the role of neutrophils and NET in atherosclerosis, however the SLE-specific mechanisms are not detailed (except at the very end of section 5.4).
Specific comments:
Line 71: a reference describing the presence of PMN or NET at sites of inflammation in diseased artery would be helpful.
Reference was added. It is reference 29, Please refer to line 89
Line 72: references are needed.
Reference was added. It is reference 30, Please refer to line 90
Line 74 : PMN are usually described as short-lived cells. However, recent data show they survive longer than believed. Activated PMN show a longer survival and PMN life span in vivo reaches 5 days in mice (Pillay et al., Blood, 2010, 116:625).
Thanks, we updated the text and reference was added. Please refer to line 93
Line 79: different forms of NET formation have been described. Some do not lead to PMN death. Therefore, NET formation is not always associated to cell death which is then named NETosis.
We have updated the text accordingly, please refer to line 100.
Line 80: Likewise, some reports suggest that some forms of NET formation do not require PAD4-mediated citrullination.
We have updated the sentence to highlight the aforementioned fact. Please refer to line 102.
Line 85: not only "double-stranded DNA" is decorated and released, as histones are also present in NET, therefore “chromatin” should be mentioned.
We have updated the sentence accordingly. Please see line 106
Line 86: Likewise “chromatin” and not “DNA strands” entrap microbes.
We have updated the sentence accordingly. Please see line 107
Line 88: it should be indicated that mitochondrial DNA may only be present in a particular mode of NET formation.
We have updated the sentence accordingly. Please see line 109
Line 94: reference 27 does not demonstrate enhanced NET formation and impaired clearance in SLE. The reference should be changed or the sentence removed.
We have updated the reference. Thanks
Line 95: it should be mentioned that the role of classical NET in SLE has been challenged (Gordon et al., JCI Insight, 2017, 2(10):e92926).
We have updated accordingly.
Paragraphs 1 and 3 of section 3 could be more detailed.
Thanks for your comment
In paragraph 3 of section 3: I would mention that neutrophils produce IFN-alpha, a key cytokine in SLE (Lindau et al., 2014, Ann Rheum Dis, 73(12):2199); that there is an impaired clearance of NET in SLE (Hakkim et al., PNAS, 2010, 107(21):9813), probably as a result of low DNase 1 activity, which is supported by data in DNase1-KO mice developing a lupus-like disease (Napirei et al., Nature genetics, 2000, 25:177 and recently confirmed by Kenny et al., Eur J Immunol, 2019, 49:590); and that, however, NET may also have immuno-regulatory properties (Ribon et al., 2019, J Autoimmun, 98:122; Kienhöfer et al., JCI Insight, 2017, 2(10):e92920).
Section has been updated
There is no section 4.
It was a typo, it has been updated. Thanks
Line 104: CCL5 is C-C motif chemokine ligand 5 (“ligand” is missing).
We added the missing word.
Line 112: remove “are”
Thanks
Line 115: thrombus NET formation should be explained.
We updated accordingly.
Lines 119 and 121: reference 40 only shows co-staining of DNA and neutrophil elastase (NE), while there is no DNA staining shown in reference 41. Therefore I would be cautious in concluding the structures are NET, especially classical NET. It can be referred to DNA-NE or MPO-citrullinated H3 complexes.
Thanks for the clarification
Line 154: the meaning of the sentence is not clear according to results described above.
We have re-worded.
Line 167: the sentence is confusing, I would remove “lesional”.
We updated accordingly.
Section 5.3: I would modulate conclusions according to the challenged role of PAD4 in lupus (Gordon et al., JCI Insight, 2017, 2(10):e92926) and the potential anti-inflammatory role of NET in lupus (Kienhöfer et al., JCI Insight, 2017, 2(10):e92920).
Section 5.3: it is not explained whether all these PAD4-mediated effects are related to NET formation.
Thanks for your comment
Line 198: the sentence is not clear.
It was re-worded
Section 5.4: mitochondrial DNA involvement is mentioned but not detailed.
We updated it accordingly.
Section 5.5: It should be explained that when studies aim at analyzing circulating NET, measuring circulating DNA or circulating nucleosomes is not enough to prove these components are NET-derived. This is often confusing in the literature and leads to over-interpretation.
We included a statement. Please refer to line 292.
Round 2
Reviewer 1 Report
The authors have appropriately answered to all my concerns.